# Public Awareness of Tuberculosis in Southeast China: A Population-Based Study

**DOI:** 10.3390/ijerph16214290

**Published:** 2019-11-05

**Authors:** Xinyi Chen, Wei Wang, Xiaomeng Wang, Chengliang Chai, Kui Liu, Ying Peng, Fei Wang, Bin Chen, Jianmin Jiang

**Affiliations:** 1School of Medicine, Ningbo University, Ningbo 315211, China; 2Department of Tuberculosis Control and Prevention, Zhejiang provincial Center for Disease Control and Prevention, Hangzhou 310051, China; jfwwang@cdc.zj.cn (W.W.); xmwang@cdc.zj.cn (X.W.); chlchai@cdc.zj.cn (C.C.); kliu@cdc.zj.cn (K.L.); ypeng@cdc.zj.cn (Y.P.); 3School of Public Health, Fudan University, Shanghai 200032, China; 4Key Laboratory of Vaccine, Prevention and Control of Infectious Disease of Zhejiang Province, Hangzhou 310051, China

**Keywords:** tuberculosis (TB), public awareness, TB control, public health

## Abstract

Few provinces in China have recently conducted population-based surveys on tuberculosis (TB) awareness at the provincial level. Hence, we conducted a population-based, cross-sectional study to evaluate the level of awareness of TB knowledge among residents of Zhejiang Province, China from October 2018 to December 2018. A total of 7174 individuals were randomly selected to participate in this survey. The rate of awareness of key information on TB was found to be 48.0%. The study’s participants exhibited a good understanding of the transmission route (80.8%), curable outcome (78.3%), and designated treatment sites (67.0%) of TB. The rate of awareness of suspicious TB symptoms (36.1%) and the relief policy on diagnosis and first-line therapeutic drugs (38.0%) were found to be relatively low among the respondents. People living in rural areas, those who were less educated, and students all showed a low level of awareness of key knowledge about TB. In conclusion, residents in Zhejiang Province generally lacked key information about TB, which is not conducive to the early detection and treatment of TB. Corresponding efforts should be made for different groups of people to achieve favorable effects on the prevention and control of TB.

## 1. Introduction

Tuberculosis (TB), an airborne infectious disease that is caused by Mycobacterium tuberculosis (MTB), remains a global human health problem. The World Health Organization (WHO)’s 2018 global tuberculosis report states that there were approximately 10.0 million new cases of TB in 2017 [1]. China, as well as 29 other countries in the WHO’s list of 30 high-TB-burden countries accounted for 87% of cases worldwide, of which China possessed 9% [1].

TB, as an ancient disease, is a leading cause of death among people in the most economically productive age groups and those living with human immunodeficiency virus (HIV) [2]. Though it is curable, patients who suffer from TB may be left with lifelong sequelae that lower their quality of life [3]. Therefore, efficient assessment, robust monitoring, and practical strategies urgently need to be implemented. According to the Sustainable Development Goal (SDGs), an annual reduction in global TB incidence of 20% by 2020 has been set as one of the milestones to achieve the ambitious goal of eliminating TB by 2030, which requires a 90% reduction in the number of deaths due to TB and an 80% reduction in global TB incidence by 2030 compared with 2015 levels [1]. In China, the Chinese Ministry of Health has launched a series of highly targeted plans to relieve the TB burden. The goal of the latest National Guideline aims to further reduce the incidence below 58/100,000 nationwide and improve the public’s awareness of key knowledge about TB prevention and control to 85% by 2020 [4].

The development of TB is more easily affected by social and economic factors than any other disease [5]. In addition to HIV infection, malnutrition, increased susceptibility of infants and the elderly, and poverty [1,6,7], a lack of awareness about TB is also considered to be an essential factor that increases the risk of exposure to TB [8].

Public awareness of TB plays a significant role in the prevention and control of this disease. A high level of TB knowledge is a crucial intervention that can influence the disease’s early detection in patients, reduce early transmissions, and increase a patient’s compliance with standard TB treatments in order to prevent multi-drug-resistant tuberculosis (MDR-TB) and extensive multi-drug-resistant tuberculosis (XDR-TB) [9]. In contrast, poor awareness not only leads to a delay in diagnosis and suboptimal treatments, but also negatively influences the social relations and moral identity of those who are afflicted with this disease [10]. Additionally, a previous study found that public awareness was tightly associated with several individual characteristics of the population, such as age, gender, education level, and occupation [11]. For these reasons, there is an urgent need to raise the health literacy of the public with respect to TB information and to carry out prevention and control measures in different populations.

In 2015, China conducted a nationwide survey on the rate of awareness of TB, which suggested that the national TB awareness rate was well below the target of 85% [12]. Since the degree of economic development, environment, and population varied wildly among different provinces, there were also large differences in the rate of awareness of TB among them. Hence, the national results cannot simply be used as the main indicators of practice at the provincial level. However, few provinces in China have recently conducted population-based investigations on TB awareness at the provincial level. Data on the lack of regional TB awareness may also affect local government public health decision-making. Zhejiang is one of the representative provinces in southeastern China. It is a relatively well developed province and, although the incidence of TB in this province was shown to have continuously declined over the last decade [13], the rate of awareness of key knowledge about TB was 44.5% in 2014 [14]. Consequently, special attention should to be paid to public awareness of knowledge about TB in Zhejiang Province, as it may provide evidence for the prevention and control of TB at the next stage of the plan.

This study aims to evaluate the knowledge and awareness that residents living in Zhejiang Province have about TB and to analyze the association of different characteristics of population with TB awareness. The findings may help to build a scientific foundation for TB awareness in Zhejiang Province and also assist public health practitioners to develop programs and strategies that more effectively serve people in southeast China.

## 2. Materials and Methods

### 2.1. Study Area and Population

The present study was conducted in Zhejiang Province, China, from October 2018 to December 2018. Geographically, Zhejiang Province consists of 11 prefectures and is located on the southeast coast of China. At the time of this survey, it had an estimated landmass area of 105,500 square kilometers and a total resident population of 57.37 million [15]. Zhejiang Province has relatively low prevalence rates in both active tuberculosis and smear-positive tuberculosis [13]. The registered incident rate was reported to be 50.8/100,000 in 2015 [16]. However, due to the large population, the burden of tuberculosis remains high in this province [17].

### 2.2. Study Design

#### 2.2.1. Sampling and Participants

The required sample size was calculated using methods that were appropriate for determination of the sample size for estimating a single proportion [18]. We assumed a significance level of 0.05, an allowable error of 0.04, and used the following equation:
(1)$$n=\frac{pq}{{\left(\frac{d}{z_{\alpha }}\right)}^{2}}$$
where *n* is the required sample size. We regarded 50% as the expected proportion of participants in the study districts that had good knowledge about TB. A non-response rate of 10% was also taken into consideration. A multistage stratified cluster sampling method was applied in our study. Firstly, in accordance with the type of economic and health development and the population size, 11 prefectures were divided into five socioeconomic categories. We then selected 11 counties to ensure that there was at least one sample county from each prefecture and 2–3 counties from each socio-economical level. Secondly, two townships were randomly selected from each county that was sampled from the first stage. Thirdly, three communities or villages from each township were selected by a random cluster sampling method to conduct the household survey. Finally, we selected 100 households per community or village using a systematic random sampling technique (Figure 1). According to the sampling principles and the stratification factors, the required sample size was 6600. From all of the eligible respondents in a household, the person whose birthday was the nearest to the survey date was recruited as the respondent. If the selected individual in the given household was unavailable, then we randomly selected another one from the set of non-sampled households. If the eligible respondent refused to do the interview, we then selected a person of the same sex with a similar age (±5 years) as an alternative, until the required sample size was reached.

Individuals aged between 12 and 75 (including 12 and 75 year olds) were eligible to be recruited. Subjects were randomly selected from the set of eligible individuals, who were required to meet the following standards: (1) had a local household registration and resided in Zhejiang during the investigation, or had a foreign registered residence but had lived in Zhejiang for more than 6 months; (2) had no communication disorders and no mental illnesses; and (3) provided written informed consent (eligible individuals who were under the age of 18 were required to obtain the consent of their guardians). Individuals were excluded if they did not meet one of the above requirements.

#### 2.2.2. Data Collection and Quality Control

All of the subjects at each survey site were interviewed via a house-to-house visit by trained and qualified interviewers using unified standard electronic questionnaires, which measured four aspects: respondents’ socio-demographic characteristics, knowledge, beliefs, and practices towards key information on TB, access to TB knowledge, and understanding of TB treatment behaviors and policies. Key information about TB was assessed using the following five questions, which can be found in the National Guideline for TB Prevention and Control [4]: (1) TB can be transmitted through a close cough, sneezing, and so forth; (2) TB should be suspected if a cough or expectoration persists for more than two weeks; (3) TB should be treated in designated TB hospitals; (4) the fees for a TB diagnosis and first-line therapeutic drugs are covered by the government; and (5) when standard treatment is adhered to, most cases of TB can be cured. We set a corresponding question item and several options that included only one correct answer for each piece of key information.

The local Center for Disease Control and Prevention (CDC) staff at each investigation site were responsible for the interviews and the collection of the questionnaires. Each investigator had received provincial training and followed unified investigation guidelines to ensure the authenticity and integrity of the entire process. The electronic questionnaire was used to control the quality of the responses and to prevent the occurrence of missing items and logistical problems. The final dataset was uniformly established at the provincial level to establish a database and carry out quality verification.

#### 2.2.3. Data Analysis

After Microsoft Excel was applied to manage and edit all of the data, IBM SPSS Statistics Version 22.0 (Armonk, NY: IBM Corp, 2013) was used for statistical analysis. The overall rate of awareness of five key pieces of TB information was calculated by the number of correct answers provided by all participants divided by the total number of answers to the five questions. Descriptive analysis of the socio-demographic characteristics of the respondents and the rate of awareness of TB was carried out. Pearson’s chi-square test and the odds ratio (OR) were used to evaluate the association between socioeconomic variables and the general awareness of TB knowledge, and the differences in the number of correctly answered TB key information questions among respondents with different characteristics. Taking the awareness of key information about TB as the outcome variable (individuals who correctly answered 4–5 TB key information items were recognized as having a high level of awareness, and individuals who correctly answered 0–3 items were classified as having a low level of awareness), a multivariate logistic regression analysis was used to evaluate the effect of socioeconomic variables on the level of awareness of key information about TB. *P* < 0.05 was considered to be statistically significant.

### 2.3. Ethical Considerations

Each participant was informed about the purpose of the study, its benefits, and confidentiality and provided a signed electronic consent form before they participated in the study. The study was conducted in accordance with the Declaration of Helsinki, and the protocol was approved by the Ethics Committee of the Zhejiang Provincial Center for Disease Prevention and Control (2018-035).

## 3. Results

### 3.1. Socio-Demographic Characteristics

As shown in Table 1, a total of 7174 questionnaires were collected in this study, which included 3323 males (46.3%) and 3851 females (53.7%). The response rate was almost 98.8% (7174/7260). The average age was 48.9 years and 48.2 years for men and women, respectively. Approximately 57% of the respondents lived in the countryside. Nearly 86% of respondents were married, and farmers accounted for more than one-third of them. More than 50% of the participants had received a secondary school education or above. Medical insurance for urban and rural residents (including the New Rural Cooperative Medical Insurance, NCMS) was the main form of medical insurance for those interviewees.

### 3.2. General Awareness of TB

With respect to general TB awareness, nearly 84% of the respondents answered that they had heard of TB (Table 2). When we compared the differences in the general awareness of TB among different characteristics of the study population, the chi-square test showed that the groups of age, residence, education status, and vocational status all showed a statistically significant difference (*P* < 0.001). However, it seemed that gender was not statistically significantly related to overall TB awareness (*P* > 0.05).

### 3.3. Awareness of Key Information about TB

In this study, the total number of key information questions that the 7174 respondents correctly answered was 17,232, and the total number of answers was 35,870. Consequently, the overall rate of awareness of key information was 48.0% (17,232/35,870) (Table 3).

More than half (3874/7174) of the study participants knew at least three or more pieces of key information about TB (Table 4). Among them, people who provided four correct answers occupied the highest proportion. Only 1% (74/7174) of the participants provided no correct answers. Table 4 also shows that, with the exception of gender, all variables reflected statistically significant differences between the corresponding characteristics and the number of correct answers.

Table 5 lists the responses to each question that was related to key information about TB. Nearly 81% and 78% of study respondents knew the mode of TB’s transmission and that TB is curable if standard treatment is adhered to, respectively. However, regarding the typical symptoms of TB, only a little more than one-third of respondents responded that infection with this disease should be suspected if a cough and expectoration persist for more than two weeks. Approximately 38% of participants were aware that the fees for a TB diagnosis and first-line therapeutic drugs are covered by the government. More than two-thirds of the respondents knew that TB should be treated in a designated hospital.

### 3.4. Awareness of Key Information about TB among People with Different Characteristics

We defined the reference groups to be: male, the 12–20 year age group, urban areas, illiterate, unemployed, and medical insurance for urban personnel. However, the multivariable stepwise analysis indicated that the differences in age composition and gender had no statistical associations with a good level of awareness of key knowledge about TB. As shown in Table 6, regarding education status, respondents with higher academic qualifications tended to have an increasingly deeper understanding of TB; those who had been educated at a university were almost three times more likely to have acquired correct key information about TB than illiterates (OR = 2.79, 95% CI: (2.03–3.83). Similarly, people who resided in urban areas exhibited a higher level of awareness of key information about TB. Most notably, when compared with medical insurance for urban workers, only medical insurance for urban and rural residents (including the New Rural Cooperative Medical Insurance, NCMS) was found to have a negative association with a high level of key knowledge about TB, while other types of insurance items suggested that there was no statistical significance. In addition, people engaged with medicine (OR = 3.96, 95% CI: 2.53–6.20), teachers (OR = 1.71, 95% CI: 1.18–2.48), and farmers (OR = 1.32, 95% CI: 1.08–1.61) were more likely to have better awareness of key information about TB than the unemployed.

## 4. Discussion

For the general public, a lack of TB awareness can contribute to low TB detection rates and the interruption of treatments [19], or even delays in early TB diagnosis [20]. This population-based study showed that the overall rate of awareness of five key pieces of information about TB was only 48.0%, an increase of nearly 3.5% and 2.4% compared with findings from 2014 and 2010, respectively [14,17]. However, there remains a long way to go to reach the 85% target of the national TB control plan by 2020, suggesting that there is an urgent need for the government and responsible agencies to further promote health and health education in relation to TB among communities.

In this survey, nearly 16% of respondents had never heard about TB; these respondents were mainly concentrated in the 12–20 year old group and the over 60 age group. This represents an increase of 3.8% and a decrease of 9.3% compared with the data on adolescents (22.8%) and the elderly (32.9%), respectively, in the National Tuberculosis Awareness Survey of 2015 [12]. The proportion of young people under 21 years old who had heard about TB was higher than in Tianjin (58.9%) but lower than the rate found in Shenzhen (77.9%) [21,22]. It is worth noting that gender did not seem to be a factor that affected the overall TB awareness rate in our study, which contrasts with the conclusion drawn in some studies that the overall rate of awareness about TB in women was generally lower than that in men [16,23,24,25]. Regarding the place of residence, respondents living in urban areas were more likely to have a better awareness of TB than those residing in rural areas, which is similar to the results of other studies conducted in China [26,27]. The focus of health education in the future should remain on the countryside.

In terms of each key piece of information about TB, the study participants had the highest level of correct knowledge about TB’s transmission route, which was much better than participants either in Vietnam (62.4%) or in Inner Mongolia (63.3%) [20,23]. The fact that TB has been characterized as a curable disease was also well comprehended by respondents, whose level of awareness on this point was also much higher than that found in an American study (32.0%) and a Tanzanian study (72%) [28,29]. Similarly, the awareness rate of TB’s designated diagnosis and treatment sites exceeded 60%. The rate of correct awareness on the question of for how long should a cough and expectoration persist in order to suspect TB, nevertheless, was far from satisfactory. It is widely known that coughing and sneezing, particularly coughing, are not only the transmission modes of some airborne infectious diseases, but also constitute typical symptoms of lung diseases such as asthma, chronic obstructive pulmonary disease (COPD), and TB. Individuals who have coughed for over two weeks and are well aware of TB would seek out a medical service and start anti-tuberculosis treatments more immediately, and thus would have a shorter duration of infection than those who take no notice of, or are not bothered by, coughing [30]. However, people are not likely to identify the early symptoms of TB (cough, fever, etc.) unless serious symptoms appear (hemoptysis, weight loss) [30]. Some people think that a persistent cough is normal, or even not a potential symptom of TB. This may be one of the reasons for the low awareness rate for this piece of key information. Similarly, the policy that the fees for a TB diagnosis and first-line therapeutic drugs are covered by the government was not well understood by our respondents. Moreover, some respondents, even if they were aware of the policy, still doubted its authenticity. In fact, the free items that the government provides cover less than 40% of the total TB diagnosis and treatment cost, which means that TB patients will still bear more than 50% of the total cost [31]. The heavy economic burden on TB patients was found to be one of the primary reasons for poor compliance with TB treatments, as well as the major problem for TB control in China and other developing countries [32]. With respect to the rates of correct answers to questions about key pieces of TB information among different groups, the results showed that access to knowledge about TB was not equal among subjects with a different socioeconomic status. We found that the amount of key knowledge about TB increased as the education level increased. This result was also confirmed in several studies [21,22]. Most notable to us in this study was that, among the various walks of life, health care workers (HCWs) tended to have the highest accuracy rate with respect to TB knowledge, with teachers ranked second. Therefore, increased involvement of highly educated health practitioners in health education delivery may have an impact on raising the public’s awareness of TB in the area. Although the rate of awareness of farmers was very low, it also provided us with a positive sign: approximately a 2.3% increase compared with the latest investigation in 2014 (41.5%) [12]. Students, however, accounted for the lowest overall accuracy with respect to key pieces of information about TB, which may be partly related to the higher proportion in the low- and middle-age groups among them in this survey. Meanwhile, the propaganda and education work on TB prevention and control was mainly allocated to TB patients and adults in society, which was also one of the reasons to explain the low awareness of students. We believed that the low awareness of TB may be one of the reasons for patients’ care-seeking delay, which may cause the cluster infection among congregate settings such as schools. As a high-risk group for clustered TB, once it breaks out, the incidence rate of students tends to exceed the average prevalence level of the local population [33], as they may not seek out immediate medical help because of their low awareness of TB. This evidence indicates that students, especially primary and secondary school ones, should definitely be the focus of the main intervention in the next phase by more novel health educational methods.

Differences reflected by careers may partially result from the relativity between the high level of awareness and the educational state of each occupational group, or it may be that occupational differences lead to unequal opportunities to acquire information about TB [22]. These two factors interacted with each other and jointly promote the improvement of the awareness rate. Generally, HCWs are considered to have a higher risk for TB infection because they are more frequently exposed to patients with active TB in the process of diagnosis, treatment, and other medical practices compared with individuals in other occupations, even with those of similar socioeconomic status, such as teachers [34,35]. The TB control plan needs to strictly implement health education on TB in order to reduce TB infection in Chinese hospitals.

In Zhejiang Province, the medical insurance system for urban workers and the NCMS constitute the main foundation of the medical security system, which covers urban and rural employees and rural residents, respectively [36]. Due to the differences in social-economic status, differences will also exist in the level of understanding of TB. A study conducted among Chinese adults suggested that medical insurance for urban workers had the greater impact on improving health-related behaviors than the NCMS [37]. This may be the reason why people who were insured under the medical insurance for urban workers scheme had a higher rate of awareness of TB.

Our study has some limitations. Firstly, TB is a disease that is related to socioeconomic development; due to the geographical constraints and taking into account the economic conditions and low TB prevalence in Zhejiang Province, the results of our study could be taken to represent southeastern China but may not fully represent the rate of awareness of key knowledge about TB at the national level, especially in areas that are relatively poor and have a high incidence of this disease. Secondly, as this study was cross-sectional, we could not draw a causal relationship between factors and effects.

## 5. Conclusions

The rate of awareness of key information about TB among residents in Zhejiang Province was found to be low and remains far from the requirement set by the National TB Control Plan. The key TB health information items need to be modified to be more convincing and precise. Emphasis should be placed on the less educated, students, and people living in the countryside using a more targeted health promotion and education campaign.

## Figures and Tables

**Figure 1 ijerph-16-04290-f001:**
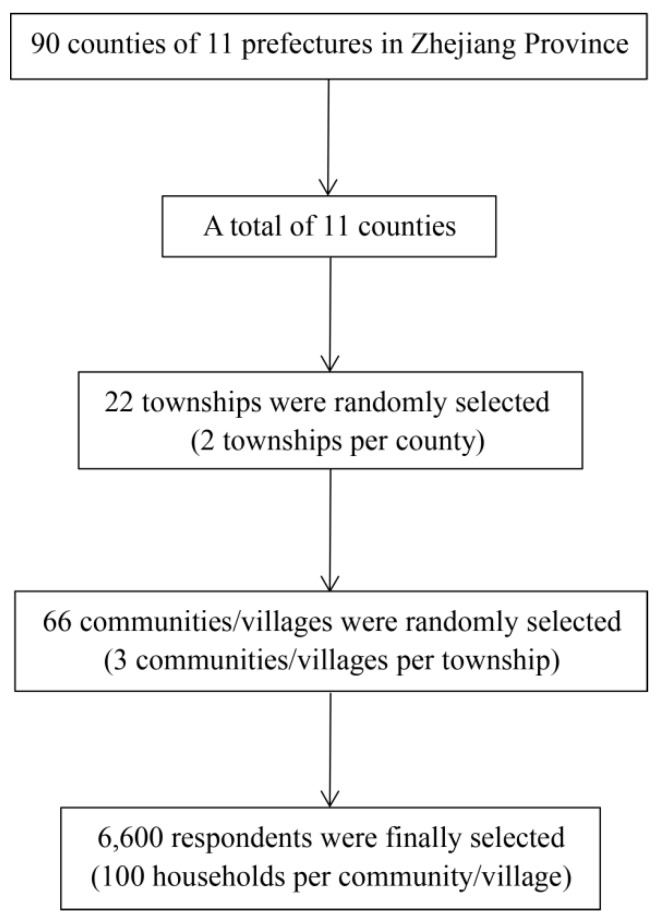
The multistage sampling process of the survey in Zhejiang Province, China from October 2018 to December 2018.

**Table 1 ijerph-16-04290-t001:** Socio-demographic characteristics of the respondents in Zhejiang Province, China from October 2018 to December, 2018.

Characteristic	Number	Percentage (%)
**Sex**		
Male	3323	46.32
Female	3851	53.68
**Age groups, years**		
12–20	307	4.28
21–30	709	9.88
31–40	1071	14.93
41–50	1531	21.34
51–60	1753	24.44
>60	1803	25.13
**Residence**		
Urban	3097	43.17
Rural	4077	56.83
**Education**		
Illiterate	817	11.39
Primary School	1786	24.9
Secondary school ^1^	2328	32.45
High school ^2^	999	13.93
Junior college	561	7.82
University	683	9.52
**Occupation**		
Government staff	323	4.5
Health care workers	153	2.13
Teachers	223	3.11
Professional technicians	585	8.15
Business/Service personnel	933	13.01
Industrial workers	960	13.38
Farmers	2679	37.34
Students	320	4.46
Unemployed	998	13.91
**Health insurance**		
Medical insurance A ^3^	2571	35.84
Medical insurance B ^4^	4317	60.18
Commercial insurance	56	0.78
Self-pay	125	1.74
Others	105	1.46
**Total**	7174	100

^1^ Secondary school: years 13–15; ^2^ High school: years 16–18; ^3^ Medical insurance A: a medical insurance system to compensate urban personnel for economic losses caused by disease risk. The reimbursement level is relatively high in China; ^4^ Medical insurance B (including The New Rural Cooperative Medical Insurance, NCMS): a medical insurance system that takes urban and rural residents who do not take part in the medical insurance scheme for urban personnel as the main objects of insurance. The reimbursement level is relatively low.

**Table 2 ijerph-16-04290-t002:** Comparison of the general tuberculosis (TB) awareness rate among respondents with different characteristics in Zhejiang Province, China from October 2018 to December 2018.

Characteristic	Heard of TB		*χ* ^2^	*P* Value
	Yes (*n*, %)	No (*n*, %)		
**Sex**			2.43	0.119
Male	2761 (83.09)	562 (16.91)		
Female	3252 (84.45)	599 (15.55)		
**Age groups, years**			177.87	<0.001
12–20	226 (73.62)	81 (26.38)		
21–30	632 (89.14)	77 (10.86)		
31–40	984 (91.88)	87 (8.12)		
41–50	1337 (87.33)	194 (12.67)		
51–60	1457 (83.11)	296 (16.89)		
>60	1377 (76.37)	426 (23.63)		
**Residence**			24.32	<0.001
Urban	2672 (86.28)	425 (13.72)		
Rural	3341 (81.95)	736 (18.05)		
**Education**			355.00	<0.001
Illiterate	550 (67.32)	267 (32.68)		
Primary School	1403 (78.56)	383 (21.44)		
Secondary school ^1^	1970 (84.62)	358 (15.38)		
High school ^2^	907 (90.79)	92 (9.21)		
Junior college	527 (93.94)	34 (6.06)		
University	656 (96.05)	27 (3.95)		
**Occupation**			221.87	<0.001
Government staff	308 (95.36)	15 (4.64)		
Health care workers	148 (96.73)	5 (3.27)		
Teachers	216 (96.86)	7 (3.14)		
Professional technicians	538 (91.97)	47 (8.03)		
Business/Service personnel	821 (88.00)	112 (12.00)		
Industrial workers	847 (88.23)	113 (11.77)		
Farmers	2102 (78.46)	577 (21.54)		
Students	240 (75.00)	80 (25.00)		
Unemployed	793 (79.46)	205 (20.54)		
**Total**	6013 (83.82)	1161 (16.18)		

^1^ Secondary school: years 13–15; ^2^ High school: years 16–18.

**Table 3 ijerph-16-04290-t003:** Rate of awareness of key knowledge about TB among respondents with different characteristics in Zhejiang Province, China from October 2018 to December 2018.

Characteristic	Total Answers (*N*)	Correct Answers (*n*)	Percentage (%)
**Sex**			
Male	16,615	7922	47.68
Female	19,255	9310	48.35
**Age groups, years**			
12–20	1535	572	37.26
21–30	3545	1789	50.47
31–40	5355	3042	56.81
41–50	7655	4039	52.76
51–60	8765	4185	47.75
>60	9015	3605	39.99
**Residence**			
Urban	15,485	7682	49.61
Rural	20,385	9313	45.69
**Education**			
Illiterate	4085	1223	29.94
Primary school	8930	3665	41.04
Secondary school ^1^	11,640	5703	48.99
High school ^2^	4995	2782	55.70
Junior college	2805	1671	59.57
University	3415	2188	64.07
**Occupation**			
Unemployed	4990	2013	40.34
Government staff	1615	1007	62.35
Health care workers	765	606	79.22
Teachers	1115	730	65.47
Professional technicians	2925	1601	54.74
Business/Service personnel	4665	2389	51.21
Industrial workers	4800	2405	50.10
Farmers	13,395	5873	43.84
Students	1600	608	38.00
**Health insurance**			
Medical insurance A ^3^	12,855	7525	58.54
Medical insurance B ^4^	21,585	9115	42.23
Commercial insurance	280	133	47.50
Self-pay	625	215	34.40
Others	525	244	46.48
**Total**	35,870	17,232	48.04

^1^ Secondary school: years 13–15; ^2^ High school: years 16–18; ^3^ Medical insurance A: a medical insurance system to compensate urban personnel for economic losses caused by disease risk. The reimbursement level is relatively high in China; ^4^ Medical insurance B (including The New Rural Cooperative Medical Insurance, NCMS): a medical insurance system that takes urban and rural residents who do not take part in the medical insurance scheme for urban personnel as the main objects of insurance. The reimbursement level is relatively low.

**Table 4 ijerph-16-04290-t004:** The number of correctly answered questions on key information about TB among respondents with different characteristics in Zhejiang Province, China from October 2018 to December 2018.

Characteristic	The Number of Correctly Answered Key Information Questions						χ^2^	*P*
	0 (n, %)	1 (n, %)	2 (n, %)	3 (n, %)	4 (n, %)	5 (n, %)		
**Sex**							5.35	0.375
Male	31 (41.89)	211 (44.23)	461 (46.24)	733 (48.16)	715 (45.69)	346 (43.96)		
Female	43 (58.11)	266 (55.77)	536 (53.76)	789 (51.84)	850 (54.31)	441 (56.04)		
**Residence**							19.11	0.002
Urban	34 (45.95)	214 (44.86)	419 (42.03)	732 (48.09)	646 (41.28)	370 (47.01)		
Rural	40 (54.05)	263 (55.14)	578 (57.97)	790 (51.91)	919 (58.72)	417 (52.99)		
**Age groups, years**							128.36	<0.001
12–20	5 (6.76)	23 (4.82)	37 (3.71)	41 (2.67)	38 (2.43)	40 (5.08)		
21–30	12 (16.22)	49 (10.27)	83 (8.32)	142 (9.33)	157 (10.03)	104 (13.21)		
31–40	9 (12.16)	61 (12.79)	147 (14.74)	261 (17.15)	286 (18.27)	152 (19.31)		
41–50	13 (17.57)	76 (15.93)	207 (20.76)	383 (25.16)	345 (22.04)	204 (25.92)		
51–60	12 (16.22)	122 (25.58)	237 (23.77)	381 (25.03)	414 (26.45)	158 (20.08)		
>60	23 (31.08)	146 (30.61)	286 (28.69)	314 (20.63)	325 (20.77)	129 (16.39)		
**Education**							255.93	<0.001
Illiterate	12 (16.22)	80 (16.77)	139 (13.94)	112 (7.36)	96 (6.13)	29 (3.68)		
Primary School	22 (29.73)	143 (29.98)	274 (27.48)	324 (21.29)	329 (21.02)	138 (17.53)		
Secondary school ^1^	21 (28.38)	139 (29.14)	317 (31.80)	537 (35.28)	521 (33.29)	247 (31.39)		
High school ^2^	8 (10.81)	63 (13.21)	117 (11.74)	250 (16.43)	265 (16.93)	135 (17.15)		
Junior college	7 (9.46)	19 (3.98)	83 (8.32)	130 (8.54)	144 (9.20)	104 (13.21)		
University	4 (5.41)	33 (6.92)	67 (6.72)	169 (11.10)	211 (13.48)	134 (17.03)		
**Occupation**							285.84	<0.001
Government staff	3 (4.05)	13 (2.73)	33 (3.31)	89 (5.85)	94 (6.01)	57 (7.24)		
Health care workers	0 (0.00)	4 (0.84)	4 (0.40)	25 (1.64)	51 (3.26)	63 (8.01)		
Teachers	1 (1.35)	4 (0.84)	28 (2.81)	50 (3.29)	80 (5.11)	40 (5.08)		
Professional technicians	6 (8.11)	53 (11.11)	65 (6.52)	153 (10.05)	126 (8.05)	91 (11.56)		
Business/Service personnel	12 (16.22)	52 (10.90)	138 (13.84)	238 (15.64)	218 (13.93)	95 (12.07)		
Industrial workers	5 (6.76)	65 (13.63)	163 (16.35)	216 (14.19)	194 (12.40)	118 (14.99)		
Farmers	25 (33.78)	173 (36.27)	364 (36.51)	513 (33.71)	597 (38.15)	209 (26.56)		
Students	8 (10.81)	21 (4.40)	43 (4.31)	41 (2.69)	47 (3.00)	38 (4.83)		
Unemployed	14 (18.92)	92 (19.29)	159 (15.95)	197 (12.94)	158 (10.10)	76 (9.66)		
**Total (N)**	74	477	997	1522	1565	787		

^1^ Secondary school: years 13–15; ^2^ High school: years 16–18.

**Table 5 ijerph-16-04290-t005:** Rate of awareness of each piece of key information about TB among the respondents in Zhejiang Province, China from October 2018 to December 2018.

Key Information Questions	Number of Respondents Who Answered (N)	Number of Respondents Who Answered Correctly (n)	Percentage (%)
TB can be transmitted through a close cough, sneezing, etc.	6013	4857	80.77
TB should be suspected if a cough and expectoration persist for more than two weeks.	6013	2168	36.06
TB should be treated in designated TB hospitals	6013	4026	66.95
The fees for a TB diagnosis and first-line therapeutic drugs are covered by the government	6013	2284	37.98
If standard treatment is adhered to, most cases of TB can be cured	6013	4708	78.30

**Table 6 ijerph-16-04290-t006:** Multivariate analysis of associated factors on the level of awareness of the five pieces of key information about TB among respondents with different characteristics in Zhejiang Province, China from October 2018 to December 2018.

Characteristic	B ^1^	S.E. ^2^	Wald ^3^	Adjusted OR ^4^ (95% CI ^5^)	*P*
**Residence**					
Urban				Reference	
Rural	−0.31	0.07	20.17	0.74 (0.64–0.84)	<0.001
**Education**					
Illiterate				Reference	
Primary school	0.43	0.12	12.70	1.54 (1.22–1.96)	<0.001
Secondary school ^6^	0.70	0.12	34.36	2.00 (1.59–2.53)	<0.001
High school ^7^	0.93	0.13	48.49	2.54 (1.95–3.30)	<0.001
Junior college	0.98	0.16	38.95	2.66 (1.96–3.62)	<0.001
University	1.02	0.16	39.88	2.79 (2.03–3.83)	<0.001
**Occupation**					
Unemployed				Reference	
Government staff	0.26	0.16	2.54	1.30 (0.94–1.78)	0.111
Health care workers	1.38	0.23	36.11	3.96 (2.53–6.20)	<0.001
Teachers	0.54	0.19	8.06	1.71 (1.18–2.48)	0.005
Professional technicians	−0.01	0.13	0.00	0.99 (0.76–1.29)	0.959
Business/Service personnel	0.04	0.12	0.09	1.04 (0.83–1.30)	0.760
Industrial workers	0.05	0.11	0.21	1.05 (0.84–1.32)	0.650
Farmers	0.28	0.10	7.34	1.32 (1.08–1.61)	0.007
Students	0.09	0.18	0.25	1.09 (0.77–1.55)	0.617
**Health insurance**					
Medical insurance A ^8^				Reference	
Medical insurance B ^9^	−0.20	0.08	7.28	0.82 (0.71–0.95)	0.007
Commercial insurance	−0.35	0.32	1.18	0.70 (0.37–1.33)	0.277
Self-pay	−0.47	0.26	3.36	0.62 (0.38–1.03)	0.067
Others	0.47	0.27	3.09	1.60 (0.95–2.70)	0.079

^1^ B, beta: regression coefficient; ^2^ S.E., standard error; ^3^ Wald, Wald χ^2^ test; ^4^ Adjusted OR, adjusted odds ratio: variables of sex and age were excluded by the logistic stepwise regression; ^5^ CI, confidence interval; ^6^ Secondary school: years 13–15; ^7^ High school: years 16–18; ^8^ Medical insurance A: a medical insurance system to compensate urban personnel for economic losses caused by disease risk. The reimbursement level is relatively high in China; ^9^ Medical insurance B (including the New Rural Cooperative Medical Insurance, NCMS): a medical insurance system that takes urban and rural residents who do not take part in the medical insurance scheme for urban personnel as the main objects of insurance. The reimbursement level is relatively low.

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
