# Peer review of "Public Awareness of Tuberculosis in Southeast China: A Population-Based Study"

_ijerph, 2019, doi:10.3390/ijerph16214290_

Round 1
Reviewer 1 Report
This is an interesting to read paper. There is clear effort in the design of the survey and its implementation. There is good descriptive and preliminary regression analysis of the data. Some comments to be considered.
Introduction
The introduction is clear. However, it has some issues that needs further consideration:
The aim of the paper is basically to evaluate the level of TB awareness in Zhejiang. However, the paper should emphasize the contribution it adds to the type of literature. From the rest of the paper, it is important to clearly indicate the reason behind choosing this particular location. It is also important that the paper highlights its contribution in terms of the survey done to implement the evaluation in hand. The paper mainly starts with stating the current status of TB and the WHO goals and China’s policy. However, the paper lacks identifying the academic contribution of the paper. There is no literature reviewed about the relationship between individual characteristics and TB awareness. This makes the paper less academic and more ‘medical report’ alike.
Methods and Results
The methods and analysis are more descriptive rather than critically analytical. The paper does not explain much of the findings, their reasoning and their implication. A reader can look at the table and just figure out its meaning. The paper should explain such tables with more analytical view. It is not clear why the authors examined the association between individual characteristics and TB awareness in two different ways as shown in table 5 and 6. These 2 tables more or less indicate similar results. Please refer to the detailed annotations on the reviewed paper.

Author Response
Response to Reviewer 1 Comments
Dear Editor and Reviewers:
We would like to take the opportunity to thank the editors and reviewers for their time and efforts. Their comments and suggestions for revision are very much appreciated. We have responded to the comments point by point. The entire article was also reviewed and modified by a professional language editor. We have highlighted the important revisions in red font in the manuscript. As there are some changes in my affiliation, I’d liked to add one affiliation to my author information and it was approved by all the co-authors. Please check the change.
Please see the comments and responses below and the revisions in the manuscript.
Regards,
Bin Chen on behalf of all the authors.
2019-10-31
Point 1: Introduction
The introduction is clear. However, it has some issues that needs further consideration:
The aim of the paper is basically to evaluate the level of TB awareness in Zhejiang. However, the paper should emphasize the contribution it adds to the type of literature. From the rest of the paper, it is important to clearly indicate the reason behind choosing this particular location. It is also important that the paper highlights its contribution in terms of the survey done to implement the evaluation in hand. The paper mainly starts with stating the current status of TB and the WHO goals and China’s policy. However, the paper lacks identifying the academic contribution of the paper. There is no literature reviewed about the relationship between individual characteristics and TB awareness. This makes the paper less academic and more ‘medical report’ alike.
Response 1: Thank you for your important comments. We have clarified the necessity for this study to be carried out in Zhejiang Province and emphasized the academic contribution to the similar type of studies (see line 64-76, line 79-81). We also have added the evidence about the relationship between TB awareness and individual characteristics (see line 59-61).
Point 2: Methods and Results
The methods and analysis are more des criptive rather than critically analytical. The paper does not explain much of the findings, their reasoning and their implication. A reader can look at the table and just figure out its meaning. The paper should explain such tables with more analytical view. It is not clear why the authors examined the association between individual characteristics and TB awareness in two different ways as shown in table 5 and 6. These 2 tables more or less indicate similar results. Please refer to the detailed annotations on the reviewed paper.
Response 2: Thank you for your important suggestions. Corresponding improvements have been modified in the expression of the results and methods respectively. We also agreed with your point that the association between the TB key knowledge and the characteristics of the population examined in two ways was redundant, so we have chosen multivariate logistic regression to evaluate it (see Table 6).

Reviewer 2 Report
General Comment:
The article addresses an important public health problem in China related to the importance of health literacy in the area of pulmonary tuberculosis.
Presents an adequate design and clear in its methodology. Their results and conclusions are consistent with the statistical analysis.
The study presents some limitations that may be important, but are mentioned by the authors in the discussion.
Line: 244, 272, 282 “Error! Bookmark not defined”.
Commentaries specifics:
Introduction: The introduction addresses the fundamentals topics.
Data analyses: Since they used multivariate analysis, the authors don't refer the co linearity effects that may exist between the variables or other confounding factors.
“the individuals who correctly answered 4 ~ 5 TB key information items were recognized as high level of awareness, and the individuals who correctly answered 0 ~ 3 items were classified as low level of awareness”. The authors should give a reason or reference for this categorization.
Results: Table 3, The information in the text is not clear in the table.
Table 4. I don't know if it is clear that: "accurate answer" corresponds to 4 or more right questions and "inaccurate answer" corresponds to 3 or less right questions? This may not be clear to the reader.
Table 5 and 6 seem redundant.
Table 6, I would say “associated factors” and not “influencing factors”
Table 6 does not contain “age” nor “residence”, any reason?
Discussion. In the discussion there is no reference to "Health insurance", given the results of tables 5 and 6, I think it deserved some comments.
Conclusion:
Conclusion on the data could be a bit more ambitious.
Author Response
Response to Reviewer 2 Comments
Dear Editor and Reviewers:
We would like to take the opportunity to thank the editors and reviewers for their time and efforts. Their comments and suggestions for revision are very much appreciated. We have responded to the comments point by point. The entire article was also reviewed and modified by a professional language editor. We have highlighted the important revisions in red font in the manuscript.
Please see the comments and responses below and the revisions in the manuscript.
Regards,
Bin Chen on behalf of all the authors.
2019-10-31
Point 1:Introduction: The introduction addresses the fundamentals topics.
Response 1: Thank you for your comment.
Point 2: Data analyses: Since they used multivariate analysis, the authors don't refer the colinearity effects that may exist between the variables or other confounding factors.
Response 2: Thank you for your comment. We’ve used the logistic stepwise regression to reduce the effect of the co-linearity of the independent variables may have on the results. We have also updated the results. (see Table 6)
Point 3: “the individuals who correctly answered 4 ~ 5 TB key information items were recognized as the high level of awareness, and the individuals who correctly answered 0 ~ 3 items were classified as low level of awareness”. The authors should give a reason or reference for this categorization.
Response 3: Thank you for your important comment. We found that the mean of the number of key information items that our respondents answered correctly was 3.18. We also have done the analysis that dividing the levels of awareness into two categories, one of which defining the respondents who correctly answered 3 ~ 5 TB key information items had a high level of awareness, and another of which defining individuals who correctly answered 0 ~ 2 items had a low level of awareness. And we have got similar results on the associate factor analysis. For these reasons, our group discussed to use the categorization presented in the text.
Point 4: Results: Table 3, The information in the text is not clear in the table.
Response 4: Thank you for your suggestion. We’ve modified the explanation of the information in the text (see lines 200-204). And we added the way to calculate the awareness rate of five key pieces of TB information in the “Data Analysis”, making it more clear. (see lines 146-148)
Point 5: Table 4. I don't know if it is clear that: "accurate answer" corresponds to 4 or more right questions and "inaccurate answer" corresponds to 3 or less right questions? This may not be clear to the reader.
Response 5: Thank you for the important comments. In fact, “accurate answer” in our study meant the correct answers to the five key information provided by all participants. We have changed it be a more appropriate expression as ‘correct answers’. We’ve modified it in Table 4 and the text. (see Table 4, lines 200-204)
Point 6: Table 5 and 6 seem redundant.Table 6, I would say “associated factors” and not “influencing factors”.Table 6 does not contain “age” nor “residence”, any reason?
Response 6: Thank you for the important comments. We admitted that Table 5 and Table 6 were a little redundant. So we deleted Table 5 since Table 6 could better describe the relationship between individual characteristics and the awareness of key knowledge. (see Table 6)
Thank you for the correction, we’ve changed the term in the title of Table 6.
Thank you for the important question. Because we used the logistic stepwise regression to analyze the associations between the different characteristics of respondents and the awareness level of key information of TB. In this way, the variables of age and gender didn’t enter the model, so we didn’t put them in Table 6. With respect to the place of residence, we were sorry to find that we had made a mistake to forget to put this variable into the model. Now we have revised all the results in Table 6.
Point 7: Discussion. In the discussion there is no reference to "Health insurance", given the results of tables 5 and 6, I think it deserved some comments.
Response 7: Thank you for your important suggestions. We’ve added more comments to the health insurance in the Discussion. (see lines 321-328)
Point 8: Conclusion on the data could be a bit more ambitious.
Response 8: Thank you for your advice. We had made some changes accordingly.
Point 9:Line: 244, 272, 282 “Error! Bookmark not defined”.
Response 9: Thank you. We’ve modified all of them, please check again. (see line 268, 300, 314)
Reviewer 3 Report
This paper describes an investigation of tuberculosis (TB) awareness among the general population of a region of southeast China. The author’s have successfully conducted a large study which undoubtedly required considerable logistical planning and oversight. Given the large number of TB cases that occur in China each year, the results are valuable for their insights into local TB awareness and knowledge.
I have a number of comments regarding the paper.
General comments:
Line 2: ‘tuberculosis’ is spelled incorrectly in the title of the paper. The manuscript requires substantial English editing, as there were sections that were not clear. A few examples that I found difficult to understand include: Sentence on lines 51 – 54. Line 54: I’m not sure what is meant by ‘cognition’ – is this referring to knowledge or understanding of TB, level of education, or something different? This terminology should be addressed throughout the paper. Line 55: ‘miss optimal treatments’ should be ‘sub-optimal treatment’. Line 67: ‘is consisted’ should be ‘consists of’; is ‘locates’ the intended word – perhaps should be ‘locales’? Line 69: ‘at that time’ – I think should be ‘at the time of this study’? Line 119: ‘logical’ I think should be ‘logistical’? Sentence on lines 250 – 252. Sentence on lines 252 – 255. There is an unnecessary level of precision in the reporting of percentages/proportions, which are stated to two decimal places throughout the paper. For example, saying that TB awareness was 48.04% (as per the abstract) is no more informative than saying that TB awareness was 48%. One decimal place (or even none) would be sufficient for percentages in this manuscript, and would give greater clarity to the results. The authors could refer to Cole 2015 Arch Dis Child 100:608-309 for a concise examination of this issue.Introduction section:
Line 37: ‘more or less’ – this term is unnecessary – if a patient has lifelong sequelae, I would think that almost certainly lowers their quality of life. Lines 39 – 41: I would prefer to see the specific target for 2030 stated, or a definition of TB elimination, rather than the general phrase ‘eliminating TB by 2030’. Lines 43 – 44: ‘etiological positive rate’ – I assume this term refers to some form of positive diagnostic result, but the wording could be clearer. For example, is it referring to confirmed active TB only; any of smear, culture, or molecular positive; or a specific gold standard? Please clarify. The introduction includes only a short paragraph about TB in Zhejiang. The authors mention the ‘12th Five-Year Plan’ and other completed indicators, but do not provide further detail. For example, what were the specific targets of the TB awareness indicator? I feel that expanding this paragraph to describe the plan and other indicators would provide stronger rationale for this paper, and how it fits into existing TB control efforts in Zhejiang.Methods section:
Lines 82 – 83: there were five socioeconomic categories, and 11 counties were selected based on these categories. Could the authors please clarify if counties from all five socioeconomic categories were represented? Lines 101 – 102: was parental consent required for children who participated in the study?Results section:
More than 25% of the participants were over 60 years of age. Could the authors please comment on how the age structure of their study participants compares to the age structure of Zhejiang more generally, and if different, how this age structure could have affected their results? Table 1: both ‘Secondary school’ and ‘High school’ are listed, but in many countries these terms are synonymous. Please provide further details in parentheses in the table to define these terms (e.g. High school (years X-X)). Table 3: how was this data statistically analysed? Was it a Chi-square test (to calculate the p-value reported on line 165)? I believe it would be informative to present confidence limits with this data. Table 5: please clarify the meaning of ‘Accurate answer’ and ‘Inaccurate answer’. It seems from the methods that these would be 4-5 correct answers or 0-3 correct answers respectively, but this is not clear from the table or legend. Table 6: why are age and residence not included in the multivariate model given they were significant in the univariate models (see Table 5)?Discussion section:
‘Medical insurance models’ are mentioned in the results (line 145) and are described in the footnotes of Tables 1, 5, and 6, but they are not mentioned in the discussion. For a reader unfamiliar with the medical insurance models used in China these results require further explanation, especially given their significance in the multivariate model (Table 6). Is there a way in which being a member of a particular insurance models could impact on a person’s TB awareness? Much of the discussion describes whether or not the present study had similar findings to other studies, which is of course important. However, I felt that the authors could have provided more detailed insights into their thoughts on why they had similar or different findings. Given the authors’ expertise in this region of China, they could also give some ideas for future research that are more specific to their results. There are a number of references that are not displaying correctly (showing as ‘Error! Bookmark not defined.’). Sentence on lines 260 – 261: is this a finding from the present study, or does it refer to other literature? If a finding of the study, the methods should state that study participants could provide extra information when they were surveyed, as this information is not presented in the results. Lines 271 – 272: the authors state that farmers showed a ‘positive signal’, but to me 44% seems quite a low level of awareness. Do they mean that it was an increase in awareness compared to a previous study? Lines 273 – 279: What was the age distribution of students in the study? Could students’ lower levels of TB awareness simply be due to their young age and relative lack of life experience, given the youngest participants in this study were only 12 years of age. In addition to low awareness among students, could higher incidence rates of TB in school outbreaks also be related to more crowded environments and the congregate setting, which may be conducive to TB transmission (see for example Andrews et al. 2014 J Infect Dis 210(4): 597-603)? The final sentence of this paragraph is unclear – are the authors suggesting that students should be a focus group for future TB awareness activities? (if so, I agree) Lines 293 – 296 – Could the authors clarify why they believe a lack of questions about MDR-TB, XDR-TB, and HIV/AIDS are a limitation of this study? These are of course important issues in relation to TB, but would awareness of DR-TB be likely to be different to awareness of DS-TB in this region? Is there a low or heavy burden of DR-TB in Zhejiang that would result in lower or higher DR-TB awareness among the population? Similarly for HIV/AIDS, is there substantial coinfection in this region, or possible misunderstanding among the population about coinfection and/or HIV? To me, it seems that survey questions about DR-TB and/or HIV would have been outside the scope of the study objective (TB awareness).Author Response
Response to Reviewer 3 Comments
Dear Editor and Reviewers:
We would like to take the opportunity to thank the editors and reviewers for their time and efforts. Their comments and suggestions for revision are very much appreciated. We have responded to the comments point by point. The entire article was also reviewed and modified by a professional language editor. We have highlighted the important revisions in red font in the manuscript.
Please see the comments and responses below and the revisions in the manuscript.
Regards,
Bin Chen on behalf of all the authors.
2019-10-31
Point 1: Line 2: ‘tuberculosis’ is spelled incorrectly in the title of the paper.
Response 1:Thank you for your correction, we’ve revised the spelling in the title. (see line 2)
Point 2: The manuscript requires substantial English editing, as there were sections that were not clear.
Response 2: According to your kindly advice, we’ve sent our manuscript to the English editor of MDPI. The editing made the description better.
Point 3: A few examples that I found difficult to understand include: Sentence on lines 51 – 54. Line 54: I’m not sure what is meant by ‘cognition’ – is this referring to knowledge or understanding of TB, level of education, or something different? This terminology should be addressed throughout the paper.
Response 3: In fact, “cognition” in this sentence referred to the awareness of TB. To avoid such similar misunderstandings, we use “awareness” in replace of “cognition” in the text. (see line 57)
Point 4: Line 55: ‘miss optimal treatments’ should be ‘sub-optimal treatment’
Response 4: Thank you for the correction, we revised it in the paper. (see line 58)
Point 5: Line 67: ‘is consisted’ should be ‘consists of’; is ‘locates’ the intended word – perhaps should be ‘locales’?
Response 5: Yes, it should be ‘consists of’, we’ve modified it. ‘locates’ was revised to ‘is located on’ for better understanding. (see line 85)
Point 6: Line 69: ‘at that time’ – I think should be ‘at the time of this study’?
Response 6: ‘at that time’ meant ‘at the time of this study’, we’ve modified it in the manuscript. (see line 86)
Point 7: Line 119: ‘logical’ I think should be ‘logistical’?
Response 7: Yes, it should be ‘logistical’, thank you for the correction. (see line 142)
Point 8: Sentence on lines 252 – 255. There is an unnecessary level of precision in the reporting of percentages/proportions, which are stated to two decimal places throughout the paper. For example, saying that TB awareness was 48.04% (as per the abstract) is no more informative than saying that TB awareness was 48%. One decimal place (or even none) would be sufficient for percentages in this manuscript and would give greater clarity to the results. The authors could refer to Cole 2015 Arch Dis Child 100:608-309 for a concise examination of this issue.
Response 8: Thank you for your important advice. With respect to precision, we’ve revised the results no more than one decimal place in both the abstract and text in order to be more concise.
Point 9: Line 37: ‘more or less’ – this term is unnecessary – if a patient has lifelong sequelae, I would think that almost certainly lowers their quality of life.
Response 9: Thank you for your suggestion. We’ve deleted the term ‘more or less’ in the introduction. (see line 39)
Point 10: Lines 39 – 41: I would prefer to see the specific target for 2030 stated, or a definition of TB elimination, rather than the general phrase ‘eliminating TB by 2030’.
Response 10: According to your advice, we’ve added more details about the phrase ‘eliminating TB by 2030’ in the introduction. (see lines 43-44)
Point 11: Lines 43 – 44: ‘etiological positive rate’ – I assume this term refers to some form of positive diagnostic result, but the wording could be clearer. For example, is it referring to confirmed active TB only; any of smear, culture, or molecular positive; or a specific gold standard? Please clarify.
Response 11: ‘etiological positive rate’ referred to the positive diagnostic results conducted by the smear, culture, or molecular. Since ‘etiological positive rate’ was not the main topic of this research and to avoid ambiguity, we decided to delete it. (see lines 46-48)
Point 12: The introduction includes only a short paragraph about TB in Zhejiang. The authors mention the ‘12th Five-Year Plan’ and other completed indicators, but do not provide further detail. For example, what were the specific targets of the TB awareness indicator? I feel that expanding this paragraph to describe the plan and other indicators would provide stronger rationale for this paper, and how it fits into existing TB control efforts in Zhejiang.
Response 12: Thank you for your important comment. The specific target of TB awareness of ‘12th Five-Year Plan’ was to raise the public’s awareness of TB key information to 85%, which we’ve mentioned in the second paragraph of introduction (see lines 47-48). We’ve added a paragraph to explain why there was an urgent need to conduct a survey about TB awareness in Zhejiang Province. (see lines 64-76)
Point 13: Lines 82 – 83: there were five socioeconomic categories, and 11 counties were selected based on these categories. Could the authors please clarify if counties from all five socioeconomic categories were represented?
Response 13: Thank you for your important comment. According to the type of economic development of Zhejiang Province, the 11 prefectures were divided into five categories. In our study, at least one county of each prefecture was randomly selected at first and thus 2-3 counties representing each socio-economical development level were included in the study to keep the representativeness. (see lines 100-103)
Point 14: Lines 101 – 102: was parental consent required for children who participated in the study?
Response 14: Yes. Because all of the subjects were interviewed via a house-to-house visit, and each of them signed the informed consent. For juveniles under 18, we also required to obtain the consent of their guardians, and we added it in the text. (see lines 122-123)
Point 15: More than 25% of the participants were over 60 years of age. Could the authors please comment on how the age structure of their study participants compares to the age structure of Zhejiang more generally, and if different, how this age structure could have affected their results?
Response 15: Thank you for the important comments. The age structure of Zhejiang Province at the time of this study was similar to the structure of our study participants. In 2018, there were nearly 23% of elderly people who were over 60 years old in Zhejiang, and in our study, it was 25%. So we think our results well represented the whole provincial level.
Point 16: Table 1: both ‘Secondary school’ and ‘High school’ are listed, but in many countries these terms are synonymous. Please provide further details in parentheses in the table to define these terms (e.g. High school (years X-X)).
Response 17: Thank you for the comment. We’ve clarified the difference between ‘secondary school’ and ‘high school’ in the footnotes below the tables. (see line 177)
Point 18; Table 3: how was this data statistically analysed? Was it a Chi-square test (to calculate the p-value reported on line 165)? I believe it would be informative to present confidence limits with this data.
Response 18: Thank you for your important comment. Chi-square test aimed to evaluate the differences between respondents with different characteristics and the number of correctly answered questions on key information about TB. We’ve provided more statistic information details in Table 4.
Point 19: Table 5: please clarify the meaning of ‘Accurate answer’ and ‘Inaccurate answer’. It seems from the methods that these would be 4-5 correct answers or 0-3 correct answers respectively, but this is not clear from the table or legend.
Response 6: Thank you for your important comment. In fact, ‘accurate answer’ referred to the correct answers to the five key information provided by all participants. We’ve applied the term of ‘correct answers’ in place of this term and added the explanation in the text. (see lines 146-148, lines 194-195 and Table 3)
Point 20: Table 6: why are age and residence not included in the multivariate model given they were significant in the univariate models (see Table 5)?
Response 20: Thank you for the important question. Because we have used the logistic stepwise regression to analyze the associations between the different characteristics of respondents and the awareness level of key information of TB. The results indicated that the variables age and gender didn’t enter the model, so we didn’t put them in Table 6. With respect to the place of residence, we were sorry to find that it was our carelessness to forget to put this variable into the model. Now we have revised all the results in Table 6.
Point 21: Medical insurance models’ are mentioned in the results (line 145) and are described in the footnotes of Tables 1, 5, and 6, but they are not mentioned in the discussion. For a reader unfamiliar with the medical insurance models used in China these results require further explanation, especially given their significance in the multivariate model (Table 6). Is there a way in which being a member of a particular insurance model could impact a person’s TB awareness?
Response 21: Thank you for the important suggestion. We’ve added some comments on the ‘Medical insurance’ in the Discussion. Because the range of people covered by different medical insurance models is different, the differences will also exist in the level of understanding of TB among them. Consequently, particular insurance models could impact on a person’s TB awareness. It was also reported in some articles. (see lines 321-328)
Point 22: Much of the discussion describes whether or not the present study had similar findings to other studies, which is of course important. However, I felt that the authors could have provided more detailed insights into their thoughts on why they had similar or different findings. Given the authors’ expertise in this region of China, they could also give some ideas for future research that are more specific to their results.
Response 22: Thank you for your important suggestion. We’ve modified our discussion.
Point 23:There are a number of references that are not displaying correctly (showing as ‘Error! Bookmark not defined.’).
Response 23: Thank you for your correction. We’ve revised all the references. (see line 268, 300, 314)
Point 24: Sentence on lines 260 – 261: is this a finding from the present study, or does it refer to other literature? If a finding of the study, the methods should state that study participants could provide extra information when they were surveyed, as this information is not presented in the results.
Response 25: Thank you for the comment. Actually, 31.84% was the proportion of the respondents who answered correctly to the question of one key information that the fees for the necessary TB diagnosis and first-line therapeutic drugs are covered by the government. In China, most of the patients are the underprivileged groups, so governments at all levels in China will provide certain fees to reduce the burden of medical treatment and improve the compliance of patients. However, not everyone knows this policy and low awareness of this information would cause the delay of seeking care, so this information has been set as one of the key information in the practice of TB control in China. And we have explained it more in detail in the result.
Point 25: Lines 271 – 272: the authors state that farmers showed a ‘positive signal’, but to me 44% seems quite a low level of awareness. Do they mean that it was an increase in awareness compared to a previous study?
Response 25: Yes, when compared to the previous study conducted in 2014, there was an increase of nearly 2.3% in the farmers’ awareness about TB key information. We’ve modified it in the Discussion. (see lines 298-299)
Point 26: Lines 273 – 279: What was the age distribution of students in the study? Could students’ lower levels of TB awareness simply be due to their young age and relative lack of life experience, given the youngest participants in this study were only 12 years of age. In addition to low awareness among students, could higher incidence rates of TB in school outbreaks also be related to more crowded environments and the congregate setting, which may be conducive to TB transmission (see for example Andrews et al. 2014 J Infect Dis 210(4): 597-603)?
Response 26: Thank you for the important comment. In our study, the age of the students was mainly concentrated between the ages of 12 and 15, of which the highest proportion was 14 years old. So the low awareness rate may partly due to their young age. Meanwhile, the propaganda and education work on TB prevention and control is mainly allocated to TB patients and adults in society, which was also one of the reasons to explain the low awareness of students. (see lines 300-304)
And we agreed with your point that the higher incidence rates of TB in school outbreaks also related to more crowded environments and the congregate setting. To avoid misunderstandings, we have revised the description in the discussion. (see lines 304-306)
Point 27: The final sentence of this paragraph is unclear – are the authors suggesting that students should be a focus group for future TB awareness activities? (if so, I agree)
Response 27: Yes, our intention was that students should be the focus group for future TB awareness activities. We’ve modified this sentence in the text. (see lines 309-311)
Point 28: Lines 293 – 296 – Could the authors clarify why they believe a lack of questions about MDR-TB, XDR-TB, and HIV/AIDS are a limitation of this study? These are of course important issues in relation to TB, but would awareness of DR-TB be likely to be different to awareness of DS-TB in this region? Is there a low or heavy burden of DR-TB in Zhejiang that would result in lower or higher DR-TB awareness among the population? Similarly for HIV/AIDS, is there substantial coinfection in this region, or possible misunderstanding among the population about coinfection and/or HIV? To me, it seems that survey questions about DR-TB and/or HIV would have been outside the scope of the study objective (TB awareness).
Response 29: Thank you for your important comment. We agree with your points and we’ve deleted this limitation in the text. (see lines 329-335)
Round 2
Reviewer 1 Report
The paper was revised more or less given the previous feedback providing a better version of it.